# Identifying agricultural disaster risk zones for future climate actions

**Nkongho Ayuketang Arreyndip** [ID] *

Department of Electrical and Electronic Engineering, University of Buea, Buea, Cameroon

* ayuketang@aims-cameroon.org

**Data Availability Statement:** Data from the Food and Agriculture Organization of the United Nations (FAO), freely available online at https://www.fao.org/faostat/en/#data/QCL.

**Funding:** The author(s) received no specific funding for this work.

## Abstract

Identifying agricultural disaster risk regions before the occurrence of climate-related disasters is critical for early mitigation planning. This paper aims to identify these regions based on data from the Food and Agriculture Organization of the United Nations (FAO), the bilateral and multilateral trade network data of the World Integrated Trade Solution(WITS) and the agent-based economic model Acclimate. By applying a uniform forcing across agricultural sectors of some breadbasket regions (US, EU and China), when single and simultaneous extreme weather events occur, such as the 2018 European heatwave, production and consumption value losses and gains are calculated at regional and global levels. Comparing the FAO data sets, WITS, and Acclimate's production value losses, the results show a strong dependence of agricultural production losses on a region's output and connectivity level in the global supply and trade network. While India, Brazil, Russia, Canada, Australia, and Iran are highly vulnerable, the imposition of export restrictions to compensate for demand shortfalls makes Sub-Saharan Africa the most vulnerable region, as it is heavily dependent on agricultural imports. In addition, simultaneous extreme weather events can exacerbate the loss of value of agricultural production relative to single extreme weather events. Agricultural practices to increase production such as smart farming, increased investment in plantation agriculture, and diversification of trading partners can help mitigate future food security risks in Sub-Saharan Africa and other agricultural import-dependent regions.

## 1 Introduction

The continuous injection of anthropogenic greenhouse gasses into the earth's atmosphere has significantly contributed to increasing global mean temperature [1–9]. This has led to the intensification and recurrent of other meteorological phenomena such as hurricanes, typhoons, extreme rainfall and floods, heatwaves. Increasing sea level rise has equally been associated with anthropogenic contribution [10–12]. Which has resulted in economic and infrastructural repercussions, with coastal cities being the hardest hit. Researchers have recently found that two or more of these extreme weather events now occur near-simultaneously in space and time. These concurrent events are becoming more frequent and growing in magnitude under increasing global mean temperature [2, 13–18].

**Competing interests:** The authors have declared that no competing interests exist.

The economic and environmental disasters of these concurrent events will therefore be more severe than the single extreme events [19]. The agricultural sector is the most vulnerable economic sector to climate change and climate-related disasters [20–22] with extreme weather events capable of significantly disrupting agricultural production. As this sector is linked to other regional economic sectors such as transport, industry, finance [23–25], shocks due to agricultural production losses can spread to other sectors thereby amplifying their overall economic impact [19]. For regions where the agricultural sector is the backbone of the economy, the impact of climate-related disasters such as heat stress-induced multiple harvest failure will gravely affect the economy [26]. As the global supply and trade network becomes increasingly complex, the economic impact of climate-related disasters can be felt in some regions far from their epicenters through the propagation of shocks down supply and trade networks. These shocks are capable of interfering with each other thereby amplifying their overall economic impact over a region [23]. Since extreme weather events are usually unpredictable, regions that experience the most impact but are not directly hit by the unprecedented event are considered here to be at risk.

The effects of extreme weather events such as heatwaves on crop yield and productivity, animal reproductivity, and their corresponding socio-economic impacts have been widely studied in the literature [27–34]. E. R. Jordan [28] investigates the effects of heat stress on reproduction. He found that, when dairy cattle are subjected to heat stress, reproductive efficiency declines. J. W. West [29] in his paper on the effects of heat-Stress on production in dairy cattle, found that increasing air temperature, temperature-humidity index, and rising rectal temperature above some critical thresholds are related to decreased dry matter intake (DMI) and milk yield and equally reduces the efficiency of milk yield. These two interesting findings in the meat and milk sectors, will lead to production shocks that can spread around the globe through supply networks. Sergei. S et al [34] investigated the effects of drought on hay and feed grain prices. By making use of an empirical example from Germany and focus on the prices of hay as well as feed wheat and barley, their results show that regional and national droughts substantially increase hay prices by up to 15%, starting with a delay of about 3 months and lasting for about a year. A thorough assessment of the evolving fragility of the global food system to price shocks was carried out by Michael J Puma et al [26]. They found a greater absolute reduction in global wheat and rice exports along with larger losses in network connectivity as the networks evolve due to disruptions in European wheat and Asian rice production. Importantly, their findings also indicate that least developed countries suffer greater import losses in more connected networks through their increased dependence on imports for staple foods. Additionally, L. Parker et al [20] investigated the vulnerability of the agricultural sector to climate change with emphasis on the development of a pan-tropical Climate Risk Vulnerability Assessment to inform sub-national decision making. The concept of agriculture losses in a telecoupled world has equally been investigated. Bren d'Amour et al [35], investigate which countries are most vulnerable to teleconnected supply shocks. They found that the Middle East is most sensitive to teleconnected supply shocks in wheat, Central America to supply shocks in maize, and Western Africa to supply shocks in rice. Vogel et al [36] had similar results when they investigated the effects of climate extremes on global agricultural yields. Connors et al [37] equally investigated agricultural losses in a telecoupled world by making use of an integrated assessment model. They demonstrated how shocks to production in one location may have profound impacts on land use and emissions in geographically distant areas.

The agent-based economic model Acclimate has widely been applied in the literature to assess first order and higher-order economic losses from natural and climate-related disasters. Wenz et al [24] made use of Acclimate to find that the increasing connectivity of international trade networks has the potential to amplify climate losses if no adaptation measures are taken

while Willner et al [25] showed that the total economic losses due to fluvial floods will increase in the next 20 years globally by 17% despite partial compensation through market adjustment within the global trade network. Kuhla et al [38] equally using Acclimate, recently showed that output losses due to heat stress alone are expected to increase by about 24% within the next 20 years if no additional adaptation measures are taken.

The concepts of agricultural losses in a telecoupled world [20, 26, 35–37, 39] and making use of Acclimate [23–25, 38] for simulating shock propagation in the global supply chain network are not new. But combining data sets from the FAO, WITS, and Acclimate to identify agricultural vulnerable regions before climate-related disasters occur is what is unique in this work. To adapt this phenomenon to a real-life scenario, the economic impacts of the 2018 European heatwave are considered. These extreme weather events in early summer 2018 were found to be connected by a recurrent hemispheric wave-7 pattern [2]. These heatwaves covered North America, Western Europe, and the Caspian Sea region, and there were also a lot of rainfall extremes in South-East Europe and Japan that occurred near-simultaneously. Researchers equally found that two or more weeks per summer spent in these waves events have been found to associate with a 4% reduction in crop production when averaged across the affected mid-latitude regions, with regional decreases of up to 11% [40]. To model the economic impact of this climate disaster scenario, a special case where the extreme weather events induce a 4% reduction in agricultural production per month spent under the extreme events when averaged over the affected breadbasket mid-latitude regions (Fig 1) is considered. To assess and identify regions at risk, the production and consumption value losses for seven different forcing scenarios including three single extreme events(USA, EU, and CHN) and four concurrent extreme events (EU-US, EU-CHN, US-CHN, and ALL (EU-US-CHN)) are computed and compared. Where USA is the single extreme event over the USA and USA-CHN represents concurrent extreme event over the USA and China. The degree of connectivity of a particular region in the global supply chain network is associated to its share of production value losses by making use of the bilateral and multilateral trade network data WITS and the

# Regions under forcing

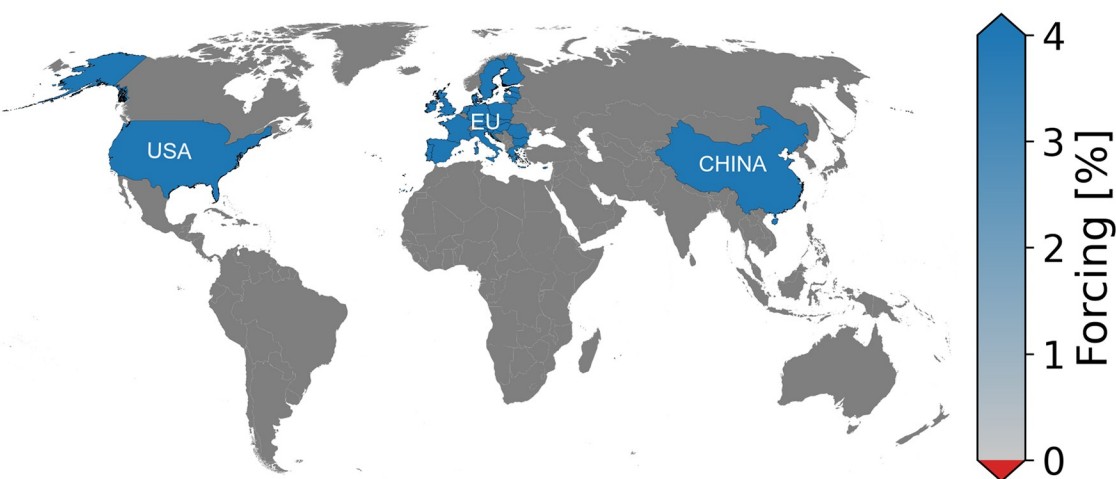

**Fig 1. The major breadbasket regions under study and regions affected by the defined events.** This figure was generated with the Mapping package in python.

FAO. This work is new as a strong dependency is found between the agricultural production value losses of a region to its connectivity in the global trade network. The rest of the section is as follows, in section 2, the source of the FAO data used, the bilateral global trade network data, the EORA economic network, and the agent-based economic model Acclimate are presented. The method of computing economic production and consumption value losses from their baseline production/consumption is presented and discussed. In section 3 and 4, the results of the numerical experiment are equally presented and discussed and a conclusion in section 5.

## 2 Materials and methods

### Data

The agricultural data set used in this study are from the Food and Agriculture Organization of the United Nations (FAO), freely available online at https://www.fao.org/faostat/en/#data/QCL. This data set covers all crops and livestock primary production quantities in tonnes for the year 2018. The year 2018 was selected to model the global impact of the 2018 heatwave. The list of Crops and Livestock primary used in this study and covered by the FAO are presented in Tables 4 and 5 in S1 File respectively while the data visualization is presented in Fig 3 in the form of a bubble map and bar charts.

The bilateral and multilateral global trade network data used are from the World Integrated Trade Solution (WITS) freely available online at WITS with the United States as the Reporter for the year 2016 (most recent on the website) aggregated over all products. Here, export is considered for trade flow at a threshold of 0.01 with the buyer as a viewpoint. The Buyer's viewpoint shows the role of each country as a source of demand in the selected sub-network. The node size (Weighted in degree) is proportional to the relevance of each country as a buyer in the selected sub-network which we also consider here as the degree of entanglement in the global supply chain network. A sample structure of the WITS network is shown in Fig 2.

### Acclimate model

Acclimate is an agent-based economic model that simulates the propagation of production losses induced by local demand, supply, or price shocks in the global supply network. Its global economy is assumed to be demand-driven with nodes in a complex network of trade and supply relations representative of firms (or regional sectors) and consumers as economic agents. Being based on local optimization principles, the model accounts for local price effects such as demand surges which are important for a comprehensive assessment of the total costs of disasters. The full description of the model is found in the paper by Otto et al [23]. This model is made up of highly interconnected regional sectors with regions representing each country in the world and the sectors are the various economic sectors that make up the economy of a country such as the agricultural sector, Food, Hotels, and Restaurants, Wholesale trade, Oil and Gas, Wood, Transport, Finance, Mining, and quarrying, etc. The economic network used in this study is the Eora26 2013 economic network which consists of 15,909 sectors across 188 countries. The multi-regional Input-Output data describe annual monetary flows between 26 major sectors and final demand in 188 countries. More about the Eora global supply chain database can be read here https://worldmrio.com/.

To simulate the spreading of economic losses caused by concurrent extreme weather events in the agricultural sector, a particular case where the extreme weather events occur near-simultaneously across very important agricultural regions of the world is assumed. The agricultural sectors of the US, EU, and China individually are shocked with a 4% forcing strength corresponding to a regional average reduction in crop production of 4% under these blocking

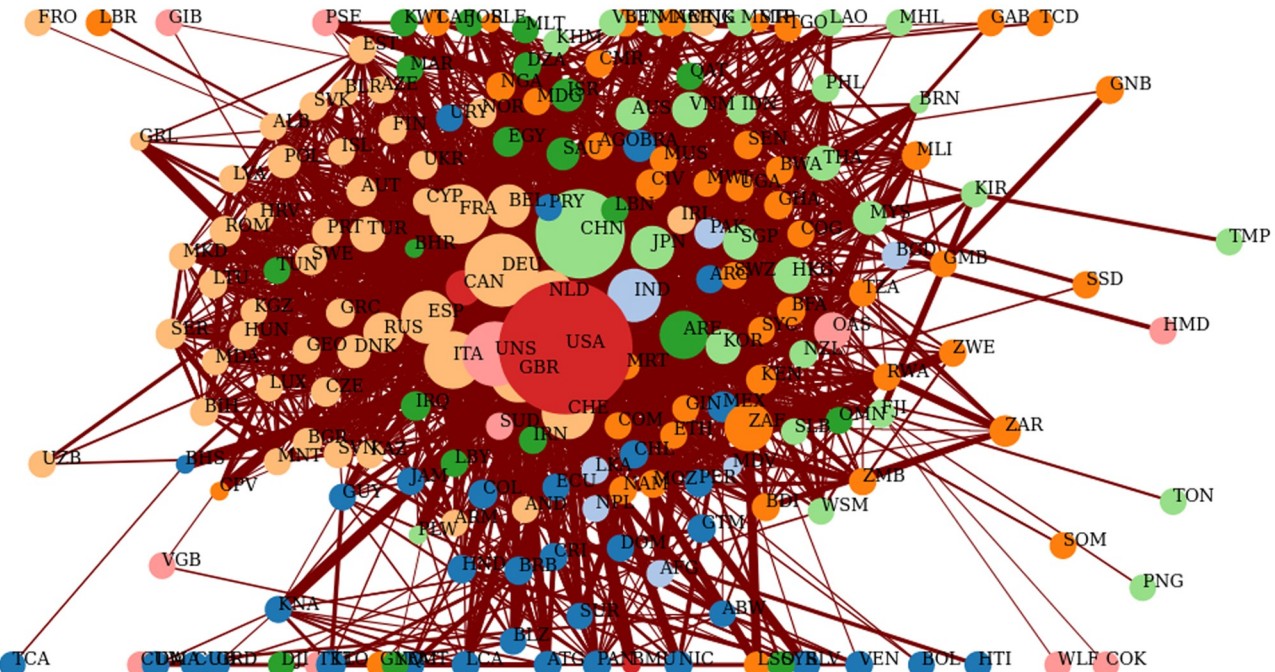

**Fig 2. A screenshot of the structure of the WITS total trade network for 2016.** Which is the most recent on the site WITS. Link thickness is proportional to country export share while the node size (Weighted in degree) is proportional to the relevance of each country as a buyer in the selected sub-network. Country names are in ISO3 format. This figure is used for the purpose of illustration.

events as reported in another research [40]. This individual forcing symbolizes the occurrence of an extreme weather event either over the US, EU, or China. Secondly, the agricultural sectors of two regions are equally simultaneously shocked such as the US and EU, US and China, and the EU and China representing the occurrence of concurrent extreme events over two breadbasket regions. Finally, the agricultural sectors of all three regions simultaneously are perturbed with same forcing strength. The simulation run time is 30 days signifying the duration of the extreme events. For each case, the direct agricultural production and consumption value losses in the directly affected regions and globally are computed. Comparative studies of the impact of each forcing scenario are also carried out. This helps in assessing which forcing has the greatest economic repercussions. To uncover which regions are more vulnerable to production value losses, the production value losses are mapped to the quantity of agricultural production and the degree of connectivity in the global supply chain network by making of the bilateral and multilateral trade network data from the World Integrated Trade Solution (WITS) and the FAO data.

### Direct economic losses

Since the agricultural sector is directly hit by extreme weather events such as heat stress and extreme precipitation, the effects of these events on crop growth and productivity, and the mental health of farmers are often severe. This sector will, therefore, experience direct economic losses such as multiple harvest failures and farmers' inefficiency due to poor mental health. These damages may also flow directly from insufficient product quality [41]. Hence, direct economic loss includes ordinary loss of bargain damages which is the difference between the actual value of the goods accepted and the value they would have had if they had been as warranted [41]. Regions that are directly hit by these disasters will experience direct

economic losses this also includes regions that import more from the affected countries than they export and produce locally such is the case in Sub-Saharan Africa while other countries will experience indirect economic losses due to trade relations with the affected countries. The total economic losses are the sum of the direct and indirect economic losses.

For each single extreme event scenario, I investigate its global economic impact by computing its production value and consumption value losses by using the expression,

$$PVL = BPV - PVF \tag{1}$$

and

$$CVL = BCV - CVF, \tag{2}$$

where PVL = Production value losses, BPV = Baseline production value, PVF = Production value under forcing, CVL = Consumption value losses, BCV = Baseline consumption value, CVF = Consumption value under forcing. We should note that, since we are computing losses, negative values imply production value under force is higher than baseline production value. Signifying a rise in production/consumption value while positive values imply a drop.

## 3 Results

Let us begin this section by looking at the top ten (10) agricultural producing regions of the world in 2018 in terms of the total crop, livestock, and total agricultural output as shown in Fig 3. Here, the share of aggregated agricultural production for the year 2018 is presented. This figure shows top agricultural producing countries. (a), (c), and (e) are bubble maps showing top crops, livestock, and total (crop + livestock) producing regions while (b), (d), and (f) are bar charts indicating the top ten crop, livestock, and total (crop + livestock) producing regions. Countries such as China, India, Brazil, the USA, Indonesia, Thailand, Russia, Nigeria, Argentina, and Vietnam lead in crop production while China, India, the USA, Brazil, Russia, Mexico, Pakistan, Japan, Germany, and Indonesia lead in livestock production. Climate-related disasters that affect any of these breadbasket regions, will have significant regional and global repercussions. To assess and identify the most vulnerable regions to climate disasters, the median production value losses when concurrent extreme weather events hit the agricultural sectors of the EU, the USA, and China are computed. Comparing these production losses to the degree of connectivity of a particular region in the global supply chain network (Table 1), we see that India with smaller production output and degree of connectivity compared to China, has a much larger share of the production losses even when it is not directly hit by the extreme weather event. This might be due to their greater import of agricultural products from these directly affected breadbasket regions than China. A similar scenario is observed between the United States and Brazil as Brazil turns to import more losses than the United States that is directly affected. Similarly, Nigeria with a much larger production output but with a smaller degree of connectivity in the global supply chain network suffers a lesser share of losses when compared to South Africa and Canada which are both highly interconnected regions. In general, this table shows that regions with a larger share of production output and a degree of connectivity above 1.0, suffer a greater share of the production losses. The case of countries such as India, Brazil, Russia showing a higher share of production value losses than the directly affected USA and China, tells us that, regions that depend more on the import of agricultural products from these directly hit breadbasket regions to meet their food demand such as the Sub-Saharan African countries are the most vulnerable regions to climate-related disasters.

The Median production and consumption value losses over all concurrent forcing scenarios are presented in Fig 4. Here, the top 10 countries with the most production value losses (a) and

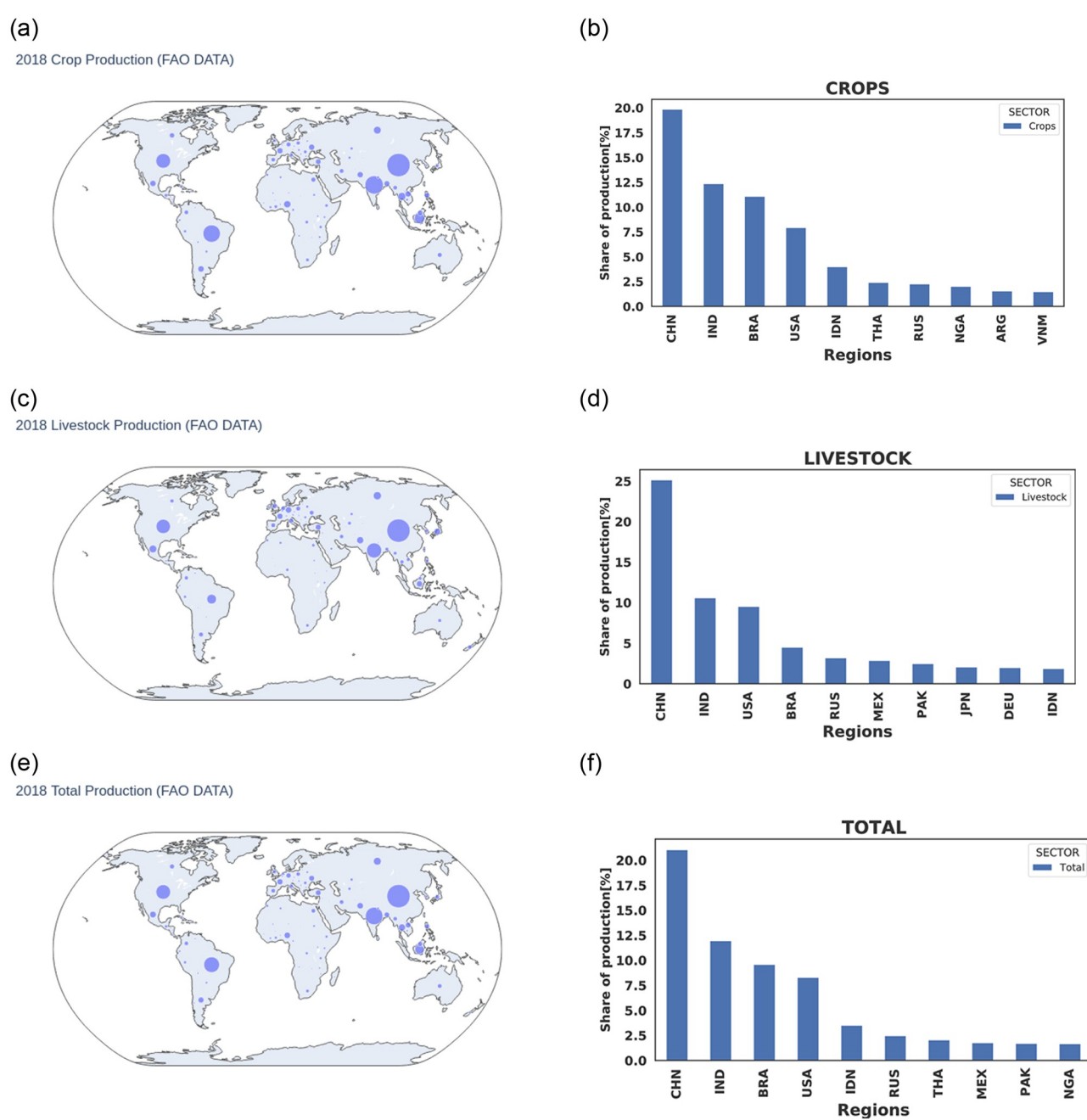

**Fig 3. Share of aggregated agricultural production for the year 2018.** This figure shows top agricultural producing countries. (a), (c), and (e) are bubble maps showing top crops, livestock, and total (crop + livestock) producing regions while (b), (d), and (f) are bar charts indicating top ten crop, livestock, and total (crop + livestock) producing regions. Names of countries in the bar charts are in ISO3 format. A list of country names, their ISO3 codes and continents can be found in Tables 6–9 in S1 File. The maps are generated using the Basemap package in python.

least production value losses/gains (b) show the most vulnerable and least vulnerable regions. Additionally, Fig 4(c) and 4(d) are top 10 consumption value losers and gainers respectively. This figure shows that India, Brazil, Russia, Canada Iran are very vulnerable regions. The USA, China, and the EU show to experience increasing in consumption value which will be transferred to consumers in the form of price hikes.

**Table 1. Comparative study to assess the effects of higher degree of entanglement in global supply and trade network and total agricultural production on cascaded agricultural production value losses from concurrent extreme weather events.**

| Regions | Total production(%) | share losses(%) | Degree of connection(Weight) |
|---|---|---|---|
| China | 21.06 | 9.14 | 10.69 |
| India | 11.96 | 15.15 | 4.57 |
| Brazil | 9.6 | 6.80 | 1.183 |
| USA | 8.3 | 1.48 | 18.28 |
| Russia | 2.47 | 5.50 | 2.32 |
| Thailand | 2.05 | 1.7 | 1.5 |
| Mexico | 1.78 | 1.82 | 0.73 |
| Pakistan | 1.7 | 0.3 | 0.69 |
| Nigeria | 1.67 | 0.2 | 0.38 |
| Canada | 1.01 | 3.29 | 1.52 |
| South Africa | 0.5 | 0.82 | 3.71 |

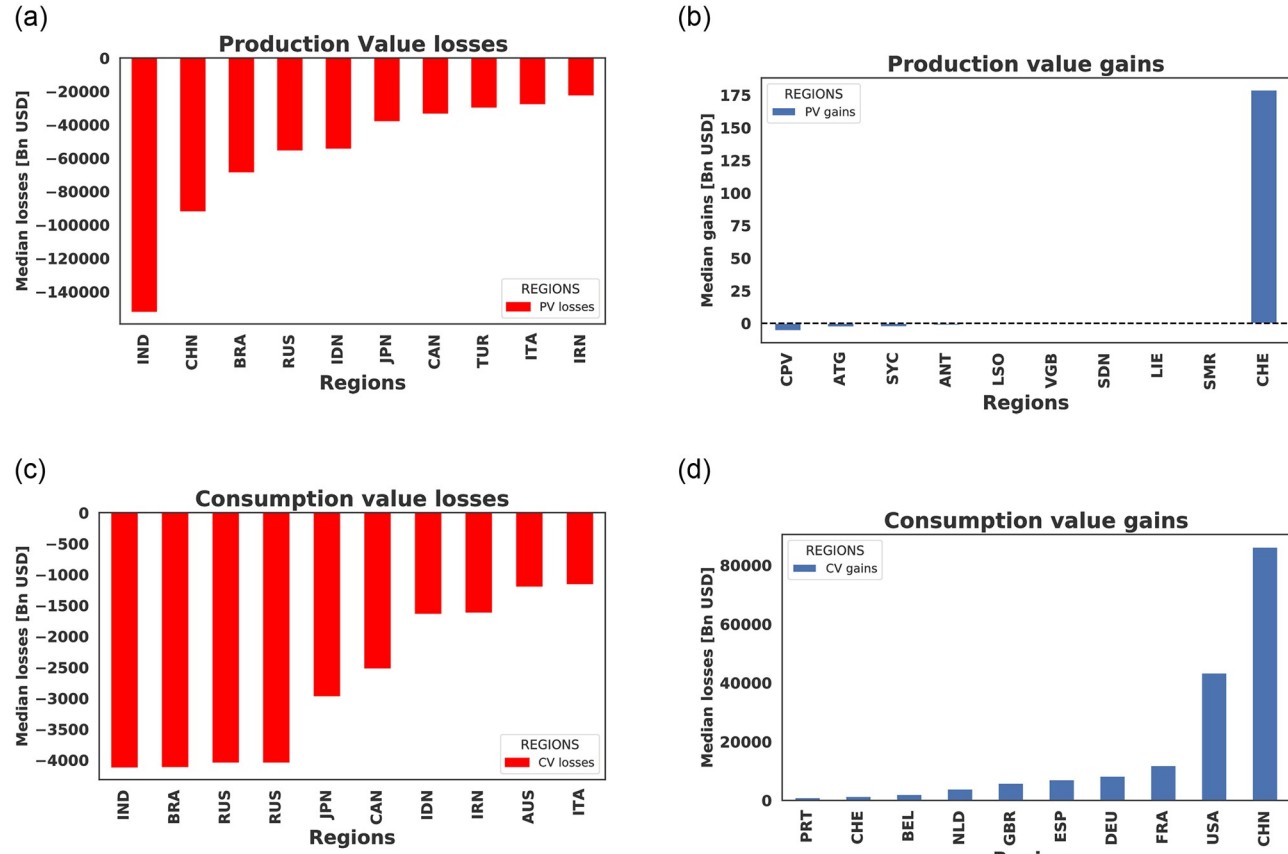

**Fig 4. Median production and consumption value losses over all concurrent forcing scenarios.** Top 10 countries with most production value losses (a) and least production value losses/gains (b). (c) and (d) are top 10 consumption value losers and gainers respectively. This figure shows that India, Brazil, Russia, Canada Iran are very vulnerable regions. The USA, China, and the EU shows price hikes.

**Table 2. Average production value and consumption value losses in the agricultural sectors for all forcing scenarios.** A comparative analysis is done over the key agricultural regions. Values are in Billion USD.

| Parameters | Key regions | EU | US | CHN | EU-US | EU-CHN | US-CHN | ALL |
|---|---|---|---|---|---|---|---|---|
| Production value losses | EU28 | 57237.3 | -31000.2 | -2807.45 | -3243.69 | -108907.88 | -245458.79 | -172373.58 |
| | USA | -24972.65 | 54086.47 | -2021.31 | 42978.49 | -119000.99 | 3864.0 | -33448.95 |
| | CHN | -49072.59 | -44116.58 | 456.63 | -142373.9 | -81471.75 | 14405.99 | -101610.68 |
| | World | -128416.55 | -103252.57 | -14997.73 | -358013.30 | -976266.56 | -936079.85 | -1222640.33 |
| Consumption value losses | EU28 | 47466.83 | -3715.05 | 36.8 | 35800.91 | 40156.09 | -21421.66 | 36247.22 |
| | USA | -4566.21 | 36355.54 | -147.0 | 40152.94 | -8811.98 | 50901.5 | 46592.79 |
| | CHN | -9125.91 | -6919.30 | 2046.11 | -10925.74 | 78680.59 | 103044.01 | 93661.27 |
| | World | 22841.38 | 26242.94 | 1812.67 | 53737.89 | 86180.55 | 89814.065 | 126860.13 |

Next, a comparative study to investigate the strength of each forcing scenario on the agricultural and economic production and consumption value is carried out. In Table 2, the average production value and consumption value losses in the agricultural sectors for all forcing scenarios (EU, US, CHN, EU-US, EU-CHN, US-CHN, and EU-US-CHN (ALL)) are presented. A comparative analysis is done over the key agricultural regions and the World. Values are in Billion USD. Here, we see that the EU agricultural sector suffers the highest production value losses when the EU is individually shocked by the extreme weather event while the highest production value rise in this region is experienced when all three regions are simultaneously forced. For the US, the US agricultural sector suffers the highest production value losses when the US is individually shocked by the extreme weather event while the highest production value rise in this region is experienced when the EU and China (EU-CHN) are simultaneously forced. For China, the Chinese agricultural sector suffers the highest production value losses when the US and China are simultaneously forced while its highest production value rise comes when the EU and the US are simultaneously forced. Globally (World), the agricultural production value rises in all forcing scenarios with the greatest rise seen when all three regions are simultaneously forced.

In this same table (Table 2), we see that the EU agricultural sector suffers the highest consumption value losses when the EU is individually shocked by the extreme weather event while the highest consumption value rise in this region is experienced when the US and China are simultaneously forced. For the US, the US agricultural sector suffers the highest consumption value losses when the US and China are simultaneously shocked by the extreme weather event while the highest consumption value rise in this region is experienced when the EU and China are simultaneously forced. For China, the Chinese agricultural sector suffers the highest consumption value losses when the US and China are simultaneously forced while its highest consumption value rise comes when the EU and the US are simultaneously forced. Globally (World), the agricultural consumption value drops in all forcing scenarios with the greatest drop seen when all three regions are simultaneously forced.

In Table 3, the cascading average production value and consumption value losses in all economic sectors for all forcing scenarios (EU, US, CHN, EU-US, EU-CHN, US-CHN, and EU-US-CHN (ALL)) are equally presented. A comparative analysis is done over the key agricultural regions and the World. Values are in Billion USD. Here, we see that the EU economy suffers the highest production value losses when the EU is individually shocked by the extreme weather event while the highest production value rise in this region is experienced when the US and China are simultaneously forced. For the US, the US economy suffers the highest production value losses when all three regions are simultaneously forced by the extreme weather event while the highest production value rise in this region is experienced when the EU and

**Table 3. Average production value and consumption value losses in all economic sectors for all forcing scenarios.** A comparative analysis is done over the key agricultural regions. Values are in Billion USD.

| Parameters | Key regions | EU | US | CHN | EU-US | EU-CHN | US-CHN | ALL |
|---|---|---|---|---|---|---|---|---|
| Production value losses | EU28 | 270176.73 | 24853.214 | 6707.73 | 213966.16 | 124159.72 | -130923.8 | 211502.13 |
| | USA | -24515.68 | 157130.23 | -288.755 | 153347.24 | -39316.55 | 239482.13 | 362005.71 |
| | CHN | -45947.04 | -33253.76 | 8520.98 | -128955.66 | 111829.99 | 337713.99 | 180484.5 |
| | World | 122660.61 | 148226.155 | 12407.3 | 74851.45 | -377620.54 | -61217.25 | 104810.96 |
| Consumption value losses | EU28 | 125557.66 | 21795.25 | 5337.77 | 115114.34 | 81388.23 | -7837.3 | 160424.93 |
| | USA | -11219.46 | 64237.42 | -102.64 | 60825.45 | -6528.27 | 114750.36 | 195370.45 |
| | CHN | -13924.12 | -15753.06 | 2734.23 | -63012.83 | 26089.3 | 154549.6 | 105844.62 |
| | World | 99748.86 | 96257.19 | 12418.95 | 83229.71 | -85313.04 | 198638.41 | 381174.07 |

China are simultaneously forced. For China, the Chinese economy suffers the highest production value losses when the US and China are simultaneously forced while its highest production value rise comes when the EU and the US are simultaneously forced. Globally (World), the global economy suffers the highest production value drop when the US alone is forced while the highest rise in production value comes when the EU and China are simultaneously forced.

In this same table (Table 2), we equally see that the EU economy suffers the highest consumption value losses when all three regions are simultaneously shocked by the extreme weather event while the highest consumption value rise in this region is experienced when the US and China are simultaneously forced. For the US, the US economy suffers the highest consumption value losses when all three regions are simultaneously shocked by the extreme weather event while the highest consumption value rise in this region is experienced when the EU alone is forced. For China, the Chinese economy suffers the highest consumption value losses when the US and China are simultaneously forced while its highest consumption value rise comes when the EU and the US are simultaneously forced. Globally (World), the global economy suffers the highest consumption value drop when all three regions are simultaneously forced while the highest rise in consumption value comes when the EU and China are simultaneously forced.

In Fig 5, a share of the median agricultural production and consumption value losses when aggregated over all concurrent extreme weather scenarios is presented. Fig 5(a) is the production value losses while Fig 5(b) is the corresponding consumption value losses. Details of the global effects of the each concurrent forcing scenario is presented in Fig 7 in S1 File. From Eqs (1) and (2), negative values(red) implies a rise in production/consumption value while positive values (blue) imply a drop in production/consumption value. Here, we see that there is a global drop in production value and a global rise in consumption value during these extreme weather events. A drop in production value will lead to losses at the level of the farmers. In other not to directly bear the losses by the farmers, these losses will be transferred to consumers in the form of higher prices (price shocks). This figure equally shows India, Brazil, China, Canada, Russia, Iran will suffer the most production value losses during concurrent extreme weather events while Brazil, India, Russia, Canada Australia, Iran, South Africa Japan, Indonesia, Argentina will generally experience price shocks.

In Fig 6, a comparative analysis of the impacts of the various forcing scenarios on the mean agricultural production Fig 6(a) and consumption values losses Fig 6(b) to test the strength of the various forcing is performed. Here, we see that concurrent extreme weather events in all three breadbasket regions will lead to the highest agricultural consumption value losses and production value gains while the least repercussion is felt when China alone is shocked.

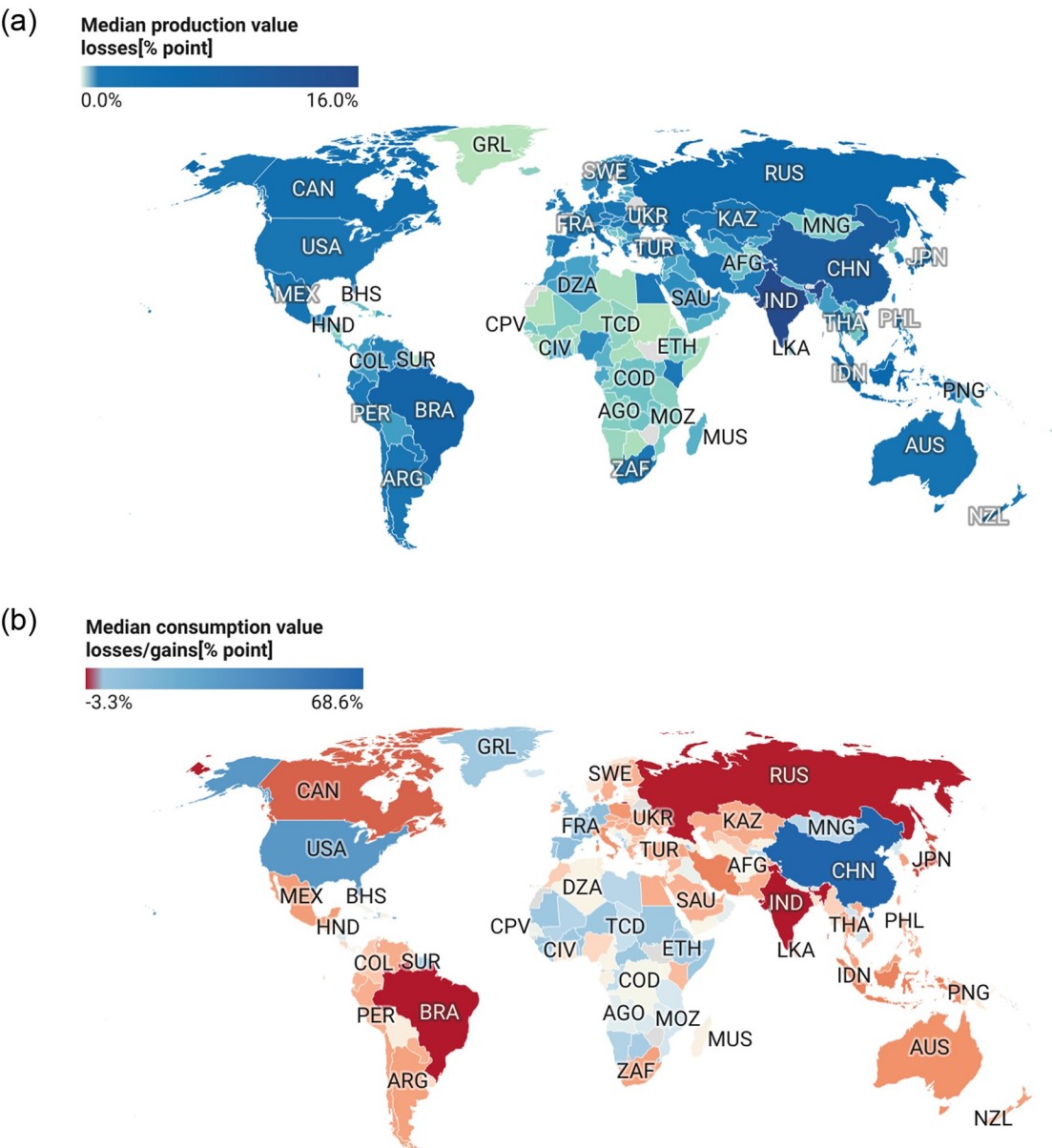

**Fig 5. A share of the median agricultural production (a) and consumption value losses (b) when aggregated over all concurrent forcing scenarios.** For the consumption value losses, negative areas experience a rise in consumption value while positive areas experience a drop in consumption value. Ones again, the bread basket regions of India, Brazil, Russia, Canada, Australia, the Middle East, Eastern Europe and most of South America, shows vulnerability. Countries names are in ISO3 format. A list of country names, their ISO3 codes and continents can be found in Tables 6–9 in S1 File Figures were generated using Datawrapper online tool.

Moreover, individual forcing over the regions leads to both the agricultural production and consumption value losses over the directly affected region. From the figure, the EU suffers the largest agricultural production value hike when the US and China are perturbed by the extreme weather events, the US greatest increase in production value is experienced when the EU and China are perturbed. While China's production value increases the most when the EU and the US are perturbed. A similar pattern is observed in the agricultural consumption values

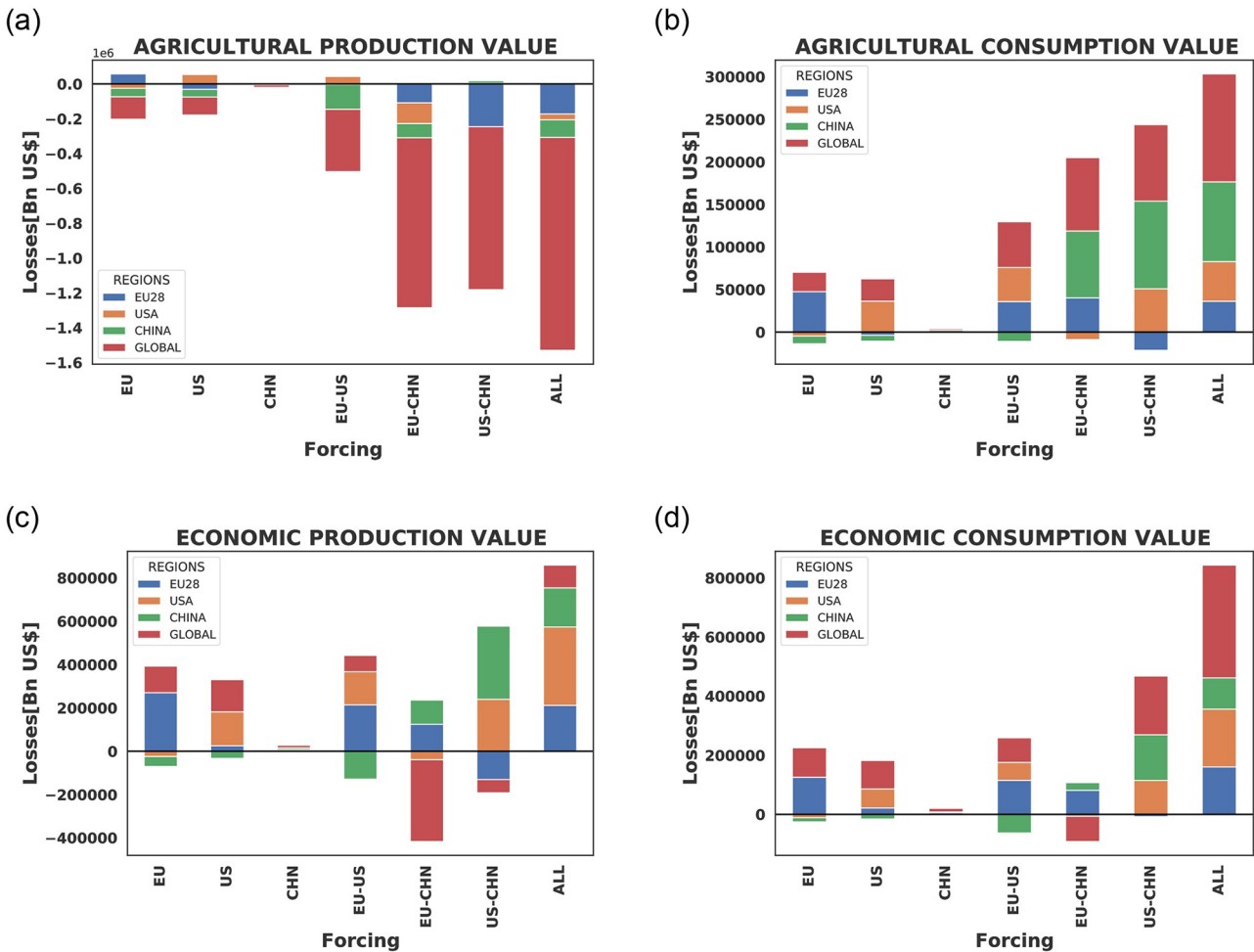

**Fig 6.** A comparative analysis of the various forcing scenarios on the mean agricultural production value losses 6a and consumption values losses 6b. Economic production value and consumption value losses are presented in 6c and in 6d respectively.

as each region experiences a rise in consumption value when it is not being affected by the blocking event and a drop in consumption value when directly perturbed.

In the entire economy (Fig 6(c) and 6(d)), we see that concurrent extreme weather events in all three breadbasket regions lead to the highest economic production and consumption values losses while the least repercussion is also felt when China alone is forced. Individual forcing over the regions equally leads to both the economic production and consumption value losses over the directly affected region. From this figure, the EU suffers the largest economic production value losses when the EU alone is shocked as compared to other concurrent event scenarios that target the EU but with the economic consumption value losses slightly higher over the EU when all three regions are perturbed. The US's greatest economic production and consumption value are experienced when all three regions are simultaneously perturbed while China suffers the highest economic production and consumption value losses when the US and China are simultaneously perturbed. A similar pattern is equally observed in the economic production and consumption value gains as each region experiences a rise in production and consumption value when it is not being affected by the blocking event.

## 4 Discussion

The agricultural sector is no doubt the most vulnerable economic sector to climate change. This is because of the direct impacts of meteorological extremes such as droughts, heatwaves, floods, extreme precipitation, and strong winds (hurricanes and tornadoes) on agricultural productivity and their disruptions to the food supply chain. While droughts exert the most impacts, heatwaves which are equally associated with increase mean temperature, have also been found to affect crop yield and productivity. Floods, extreme precipitation, hurricanes, bush fires, and insect pest will equally destroy cropland, leading to low output from farms. For regions where the economy largely depends on subsistence agriculture, the impact of climate change will be heavy, posing a threat to regional food security. Under increasing international trade linkages, climate-induced agricultural losses in one part of the world can significantly affect business in another through the propagation of shocks down trade networks. Researchers have associated increase network losses with its increasing complexity [24]. This makes highly inter-connected regions very vulnerable to economic losses. The simulations carried out in this paper aims at identifying vulnerable agricultural regions to climate change for early action to mitigate impacts such as designing a more resilient agricultural sector. The findings show that highly interconnected breadbasket regions such as India, Brazil, Russia, Canada, Australia, the Middle East, Eastern Europe, and most of South America are vulnerable to network losses with India, Brazil, and Russia showing the highest vulnerability. Very similar results have been obtained by Bren d'Amour et al [35] and Vogel et al [36] proving that indeed agricultural losses is the network and output quantity dependent. At the regional level, policies such as imposing export restrictions to compensate for demand deficits may secure regional food banks for major food-producing regions but will put the lives of millions of people in regions that import more than they produce locally. Sub-Saharan Africa is one of those regions with low agricultural output. Agricultural practices to increase production such as smart agriculture, increase investment in plantation agriculture, and diversifying regional and international trade partners, may help mitigate future food security risks in Sub-Saharan Africa.

Some of the limitations of this work are that a uniform forcing is considered over the breadbasket regions where in reality, extreme weather-induced agricultural losses aren't uniform, as many affected areas suffer different magnitudes of losses. Moreover, the FAO data used is for the year 2018 and the WITS network data is for the year 2016. We expect further research that uses the agricultural and network data of that year including data about the fraction of agricultural losses due to an extreme weather event in that year for a thorough impact assessment.

## 5 Conclusion

Early disaster warnings usually call for prompt action to mitigate impacts. Identifying agricultural disaster risk zones before climate-related disasters occur helps in designing effective disaster risk reduction strategies and policies for regional and global food security. In an increasingly inter-connected world through supply and trade networks, the economic impact of climate-related disasters can be felt in some regions far from their epicenters through the propagation of shocks down supply and trade networks. These regions are considered here to be at risk. Moreover, shocks coming from different trade routes might overlap thereby amplifying their overall economic impact over a region. The agent-based economic model Acclimate together with the EORA 2013 economic network, the FAO agricultural production data for the year 2018, and the bilateral and multilateral trade network data from the World Integrated Trade Solution(WITS) have been employed to assess and identify agricultural disaster risks zones. A uniform forcing has been applied over some breadbasket regions (USA, EU, and China) when single and concurrent extreme weather events occur such as the case of the 2018

European heatwave. The direct agricultural and economic production and consumption value losses and gains in the regional and global agricultural sectors and the entire economy for all forcing scenarios have been computed and compared. Results have shown a strong dependence of agricultural production losses on the quantity of production and the degree of connectivity of a region in the global supply and trade network. Additionally, regions with a larger share of production output and a degree of connectivity above 1.0, suffer a greater share of the production losses. Breadbasket regions such as India, Brazil, and Russia were found most vulnerable. If these regions and other breadbasket regions such as the EU, USA, and China are to impose export restrictions to compensate for demand deficits, millions of people in Sub-Saharan Africa will be at risk of starvation. This risk in future food security can be mitigated through agricultural practices to increase production such as smart agriculture, increase investment in plantation agriculture, and diversifying regional and international trade partners.

It is equally worth noting that concurrent extreme weather events show a greater impact in both the agricultural sectors and the global economy compared to a single extreme event scenario. More resilient agricultural systems are recommended to handle the impact of concurrent extreme weather events as they are likely to become more frequent and intense under increasing global mean temperature.

## Supporting information

**S1 File.**
(PDF)

## Acknowledgments

The author will like to thank the Alexander von Humbold Foundation for the financial support in the form of a stipend for the International Climate Protection Fellowship 2020. The author will also want to thank the members of the Department of Complexity Sciences of the Potsdam Institute for Climate Impact research (PIK) in Potsdam, Germany for their immense material support during the time spent there.

## Author Contributions

**Conceptualization:** Nkongho Ayuketang Arreyndip.

**Formal analysis:** Nkongho Ayuketang Arreyndip.

**Investigation:** Nkongho Ayuketang Arreyndip.

**Methodology:** Nkongho Ayuketang Arreyndip.

**Project administration:** Nkongho Ayuketang Arreyndip.

**Validation:** Nkongho Ayuketang Arreyndip.

**Visualization:** Nkongho Ayuketang Arreyndip.

**Writing – original draft:** Nkongho Ayuketang Arreyndip.

**Writing – review & editing:** Nkongho Ayuketang Arreyndip.

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
