## [Decision Letter · Decision Letter 0]

4 Jul 2021

PONE-D-21-15661

Identifying agricultural disaster risk zones for future climate actions.

PLOS ONE

Dear Dr. Arreyndip,

Thank you for submitting your manuscript to PLOS ONE. After careful consideration, we feel that it has merit but does not fully meet PLOS ONE’s publication criteria as it currently stands. Therefore, we invite you to submit a revised version of the manuscript that addresses the points raised during the review process.

We look forward to receiving your revised manuscript.

Kind regards,

Ghaffar Ali, PhD

Academic Editor

PLOS ONE

Journal Requirements:

[This research has received financial support from the Alexander von Humboldt Foundation through the International ClimateProtection (ICP) Fellowship Program 2020.The author will also want to thank the Ministry of Higher Education of Cameroon for the financial support in the form ofresearch allowances for State University lecturers.The author will also want to thank the members of the Department of Complexity Sciences of the Potsdam Institute forClimate Impact research (PIK) in Potsdam, Germany for their immense material support during the time spent there. ]

 [The author(s) received no specific funding for this work.]

3. We note that Figures 1, 2, 3 and 5 in your submission contain map images which may be copyrighted. All PLOS content is published under the Creative Commons Attribution License (CC BY 4.0), which means that the manuscript, images, and Supporting Information files will be freely available online, and any third party is permitted to access, download, copy, distribute, and use these materials in any way, even commercially, with proper attribution. For these reasons, we cannot publish previously copyrighted maps or satellite images created using proprietary data, such as Google software (Google Maps, Street View, and Earth). For more information, see our copyright guidelines: http://journals.plos.org/plosone/s/licenses-and-copyright.

You may seek permission from the original copyright holder of Figures 1, 2, 3 and 5 to publish the content specifically under the CC BY 4.0 license. 

If you are unable to obtain permission from the original copyright holder to publish these figures under the CC BY 4.0 license or if the copyright holder’s requirements are incompatible with the CC BY 4.0 license, please either i) remove the figure or ii) supply a replacement figure that complies with the CC BY 4.0 license. Please check copyright information on all replacement figures and update the figure caption with source information. If applicable, please specify in the figure caption text when a figure is similar but not identical to the original image and is therefore for illustrative purposes only.

Reviewers' comments:

Reviewer's Responses to Questions

**Comments to the Author**

1. Is the manuscript technically sound, and do the data support the conclusions?

Reviewer #1: Partly

Reviewer #2: Yes

2. Has the statistical analysis been performed appropriately and rigorously? 

Reviewer #1: Yes

Reviewer #2: Yes

3. Have the authors made all data underlying the findings in their manuscript fully available?

Reviewer #1: Yes

Reviewer #2: Yes

4. Is the manuscript presented in an intelligible fashion and written in standard English?

Reviewer #1: Yes

Reviewer #2: Yes

5. Review Comments to the Author

Reviewer #1: This manuscript claims to detail the agricultural disaster risk zones based on the data from FAO and WITS. The subject matter discussed here is relevant to the journal and is important to study as well. Overall, the paper is well structured and the presentation is good. However, due to several issues at the current moment, I would recommend returning the manuscript for some major revisions in almost each section of the submitted draft. Please see the following comments and suggestions.

Abstract:

The problem statement is taking too much space in the abstract. Try to confine it in two sentences at max. and narrate the objective, which as of now is starting after 5 lines.

Please turn the sentences to remove “I” from all spaces in the abstract and check similar throughout the manuscript.

At the moment, the abstract does not contain any concrete/solid results. Please include significant results.

Introduction:

After the 2nd paragraph, there should be details on economic impacts from agricultural sector to show the severity of the issue discussed. This can make the case stronger to conduct such studies. In its present form, the case is not strong enough. Similarly, the author needs to go through the idea of agriculture losses in a telecoupled world (see for example: https://onlinelibrary.wiley.com/doi/abs/10.1002/9781119413738.ch5) to strengthen the literature review in this field. The idea of highlighting agricultural losses and climate extremes as well as network dependence is not new (as also stated in the manuscript and can be seen at https://iopscience.iop.org/article/10.1088/1748-9326/ab154b and https://iopscience.iop.org/article/10.1088/1748-9326/ab4864 ). Therefore, it is crucial to signify the objectives and research question addressed in this manuscript, which is weak at the moment.

Data and Methods:

The data used from FAO is for 2018 and for WITS it is 2016. Wouldn't that old WITS data influence the results as the networks data might have been updated? Further, why the latest data from both of these is not used? you need to explain it in this section. The methods section is too concise and the readers are referred to links/resources to check themselves about the models used. There should be more details on these techniques and models included in the manuscript for reader’s sake. It should not be left to the readers to figure out the techniques used for the analysis as it might result in confusions.

Is extreme heat the only disaster considered here? What about floods, tropical cyclones/hurricanes etc., which are one of the most significant natural hazards in the context of agricultural losses in China and USA?

Results and Discussion:

This section is primarily focused on results only and no discussion is provided to show the usefulness of the findings from this study. The points highlighted in the introduction section should be revisited in the discussion section to show how the study would be significant in providing solutions to the issues raised (complete or partial solutions). Similarly, thoughts should also be given to the applicability of the results in terms of disaster risk reduction related policy—even though the disaster considered here is merely covers a smaller proportion of agricultural impacts as compared to other significant disasters such as floods and typhoons. This further highlighted the need of representing agricultural impacts due to different disasters in the introduction section as suggested above. It should be thoroughly discussed that what are the implication of these findings at national and sub-national especially local levels. Which results among the provided figures detail risk zones? You need to explicitly present the results of zones as it is reflected directly from the title of the manuscript. What are the key zones among those? How this zoning can effectively be utilized for adaptation and better decisions as well as resource allocation to reduce the possible impacts in the face of climate change? Furthermore, what are the current limitations and future prospects? How this study could pave ways for further research in this field? All these are missing from the results and discussion section.

Finally, the conclusions section seems like a replicate of abstract. Therefore, it should be revised carefully to represent significant conclusions based on a deep thinking of results and discussion. Not just repeat the already detailed things.

Lastly, the manuscript also needs a low to intermediate revision regarding grammatical and English language (i.e., 2nd sentence in introduction, 1st sentence in Section 2.1). “Data” is plural so you must use “are” and not “is”. Sometimes, the sentences are too long to follow (e.g., first two sentences of 1st paragraph on page 2).

Reviewer #2: This is an interesting article presenting methodology to identify agricultural disaster risk zones which is vital to combat possible climate change impacts in future. The article is well written, it can be considered for publication after addressing the following comments.

Abstract: Well written – no comments.

Introduction: The introduction is very good, the authors demonstrate a thorough knowledge of the published literature and highlight the importance and background to carry out this investigation

Methods: Methods are technically strong and well explained.

Results and Discussion: Results are well explained. However, the discussion concerning other published papers on the topic must be included.

Fig 2, The legend in the sub-figures do not make much sense. The author may try to elaborate on the regions in fig 2(a) and the abbreviations in figures 2b-d.

Conclusion: No Comment.

6. PLOS authors have the option to publish the peer review history of their article (what does this mean?). If published, this will include your full peer review and any attached files.

Reviewer #1: No

Reviewer #2: No

---

## [Author Response · Author response to Decision Letter 0]

6 Oct 2021

Response to Reviewer 1

Reviewer #1: This manuscript claims to detail the agricultural disaster risk zones based on the data from FAO and WITS. The subject matter discussed here is relevant to the journal and is important to study as well. Overall, the paper is well structured and the presentation is good. However, due to several issues at the current moment, I would recommend returning the manuscript for some major revisions in almost each section of the submitted draft. Please see the following comments and suggestions.

Response: Thank you very much for your complements and suggestions. Your suggestions have been carefully taken into consideration and have significantly contributed to improving the content of the manuscript.

Abstract:

The problem statement is taking too much space in the abstract. Try to confine it in two sentences at max. and narrate the objective, which as of now is starting after 5 lines. Please turn the sentences to remove “I” from all spaces in the abstract and check similar throughout the manuscript. At the moment, the abstract does not contain any concrete/solid results. Please include significant results.

Response: The problem statement has been restructured. The entire manuscript has been turn to remove ‘I’ as recommended. Concrete results have been included in the abstract. The entire abstract has been restructured. Thank you.

Introduction:

After the 2nd paragraph, there should be details on economic impacts from agricultural sector to show the severity of the issue discussed. This can make the case stronger to conduct such studies. In its present form, the case is not strong enough. Similarly, the author needs to go through the idea of agriculture losses in a telecoupled world (see for example: https://onlinelibrary.wiley.) to strengthen the literature review in this field. The idea of highlighting agricultural losses and climate extremes as well as network dependence is not new (as also stated in the manuscript and can be seen at https://iopscience.iop.org/ and https://iopscience.iop.org/ ). Therefore, it is crucial to signify the objectives and research question addressed in this manuscript, which is weak at the moment.

Response: Related literature has been added to make the case stronger as suggested. The originality of the work has been highlighted in the last paragraph of the introduction.

Data and Methods:

The data used from FAO is for 2018 and for WITS it is 2016. Wouldn't that old WITS data influence the results as the networks data might have been updated? Further, why the latest data from both of these is not used? you need to explain it in this section. The methods section is too concise and the readers are referred to links/resources to check themselves about the models used. There should be more details on these techniques and models included in the manuscript for reader’s sake. It should not be left to the readers to figure out the techniques used for the analysis as it might result in confusions.

Is extreme heat the only disaster considered here? What about floods, tropical cyclones/hurricanes etc., which are one of the most significant natural hazards in the context of agricultural losses in China and USA?.

Response: The reason why the FAO data for 2018 was used has been included and the changes highlighted. The WITS data of 2016 is the most recent. There is no other new data on the website. This reason has also been included in the Data and Method section. 

Literature about the use of the agent-based economic model Acclimate to assess economic impacts of other disasters and policies have been included. The algorithm of the model Acclimate is very complex and has been described and published by the original developer in the paper by Otto et al. Every other paper that uses Acclimate like this one, just explain how the model functions while referencing Otto et al paper. So details about the model is beyond the objectives of this paper. A structure of the WITS network has been included in Figure 2 for clarifications and the link to the page has been provided. The FAO data has equally been referenced. 

Data from the Nature Climate Change paper Kai et al (2018) was used as forcing data. Rossby waves that generate this forcing data covers heatwaves and floods. While the USA and the EU experience heatwaves, China experiences floods especially South East of China. On average, Kai et al (2018) found a 4% reduction in crop production when averaged over these regions. This is what was considered for shocking into the Acclimate.

Results and Discussion:

This section is primarily focused on results only and no discussion is provided to show the usefulness of the findings from this study. The points highlighted in the introduction section should be revisited in the discussion section to show how the study would be significant in providing solutions to the issues raised (complete or partial solutions). Similarly, thoughts should also be given to the applicability of the results in terms of disaster risk reduction related policy—even though the disaster considered here is merely covers a smaller proportion of agricultural impacts as compared to other significant disasters such as floods and typhoons. This further highlighted the need of representing agricultural impacts due to different disasters in the introduction section as suggested above. It should be thoroughly discussed that what are the implication of these findings at national and sub-national especially local levels. Which results among the provided figures detail risk zones? You need to explicitly present the results of zones as it is reflected directly from the title of the manuscript. What are the key zones among those? How this zoning can effectively be utilized for adaptation and better decisions as well as resource allocation to reduce the possible impacts in the face of climate change? Furthermore, what are the current limitations and future prospects? How this study could pave ways for further research in this field? All these are missing from the results and discussion section.

Response: The discussion section has been added and the usefulness of the findings have been briefly discussed. Figures 4 and 5 details the risk zones and this has been discussed in the text.

Current limitations and future prospects paragraph has been added in the last paragraph of the discussion section.

Finally, the conclusions section seems like a replicate of abstract. Therefore, it should be revised carefully to represent significant conclusions based on a deep thinking of results and discussion. Not just repeat the already detailed things.

Response: This section has been restructured to represent a summary of all what has been done and the relevance.

Lastly, the manuscript also needs a low to intermediate revision regarding grammatical and English language (i.e., 2nd sentence in introduction, 1st sentence in Section 2.1). “Data” is plural so you must use “are” and not “is”. Sometimes, the sentences are too long to follow (e.g., first two sentences of 1st paragraph on page 2).

Response: Corrections have been made and grammar has been revisited to the best of my knowledge. Thank you ones again. 

Response to Reviewer 2

This is an interesting article presenting methodology to identify agricultural disaster risk zones which is vital to combat possible climate change impacts in future. The article is well written, it can be considered for publication after addressing the following comments.

Response: Thank you very much for your complements and your suggestions have helped to improve the content of the paper.

Abstract: Well written – no comments.

Response: The abstract has been restructured for a better understanding.

Introduction: The introduction is very good, the authors demonstrate a thorough knowledge of the published literature and highlight the importance and background to carry out this investigation

Response: Thank you.

Methods: Methods are technically strong and well explained.

Response: Thank you.

Results and Discussion: Results are well explained. However, the discussion concerning other published papers on the topic must be included.

Response: A new section on discussion has been added and the discussions about how the results here tie to other results have been included. Thank you.

Fig 2, The legend in the sub-figures do not make much sense. The author may try to elaborate on the regions in fig 2(a) and the abbreviations in figures 2b-d.

Response: The legends were wrong and have been updated. New figures have been added here for clarification. The ISO3 codes were used in the place of country names. A list of country names and their ISO3 codes have been added in the supplementary section for a better comprehension.

Conclusion: No Comment.

Response: This section has equally been updated. Thank you.

---

## [Decision Letter · Decision Letter 1]

10 Nov 2021

Identifying agricultural disaster risk zones for future climate actions.

PONE-D-21-15661R1

Dear Dr. Arreyndip,

We’re pleased to inform you that your manuscript has been judged scientifically suitable for publication and will be formally accepted for publication once it meets all outstanding technical requirements.

Kind regards,

Ghaffar Ali, PhD

Academic Editor

PLOS ONE

Additional Editor Comments (optional):

Reviewers' comments:

Reviewer's Responses to Questions

**Comments to the Author**

1. If the authors have adequately addressed your comments raised in a previous round of review and you feel that this manuscript is now acceptable for publication, you may indicate that here to bypass the “Comments to the Author” section, enter your conflict of interest statement in the “Confidential to Editor” section, and submit your "Accept" recommendation.

Reviewer #1: All comments have been addressed

Reviewer #2: All comments have been addressed

2. Is the manuscript technically sound, and do the data support the conclusions?

Reviewer #1: Yes

Reviewer #2: Yes

3. Has the statistical analysis been performed appropriately and rigorously? 

Reviewer #1: Yes

Reviewer #2: Yes

4. Have the authors made all data underlying the findings in their manuscript fully available?

Reviewer #1: Yes

Reviewer #2: Yes

5. Is the manuscript presented in an intelligible fashion and written in standard English?

Reviewer #1: Yes

Reviewer #2: Yes

6. Review Comments to the Author

Reviewer #1: All the comments raised in the previous round are addressed thoroughly, and the author has revised the manuscript intensively. I recommend acceptance of the manuscript after formal editorial procedures.

Reviewer #2: Authors have revised the manuscript according the comments raised. I am fine with the revised version.

7. PLOS authors have the option to publish the peer review history of their article (what does this mean?). If published, this will include your full peer review and any attached files.

Reviewer #1: No

Reviewer #2: No

---

## [Editor Report · Acceptance letter]

17 Nov 2021

PONE-D-21-15661R1 

Identifying agricultural disaster risk zones for future climate actions 

Dear Dr. Arreyndip:

I'm pleased to inform you that your manuscript has been deemed suitable for publication in PLOS ONE. Congratulations! Your manuscript is now with our production department. 

Kind regards, 

on behalf of

Prof. Ghaffar Ali 

Academic Editor

PLOS ONE